# Increasing the willingness to participate in organ donation through humorous health communication: (Quasi-) experimental evidence

Cornelia Betsch[1]*, Nora K. Schmid-Küpke[2], Leonie Otten[1], Eckart von Hirschhausen[3]

1 Health Communication, Media and Communication Science, Center for Empirical Research in Economics and Behavioral Sciences, University of Erfurt, Erfurt, Germany, 2 Department for Infectious Disease Epidemiology, Immunization Unit, Robert Koch Institute, Berlin, Germany, 3 Humor hilft Heilen Foundation, Bonn, Germany

* cornelia.betsch@uni-erfurt.de

**Data Availability Statement:** The data that support this study's findings are openly available in the Open Science Framework repository at https://osf.io/vkn4s/.

## Abstract

Increasing people's willingness to donate organs after their death requires effective communication strategies. In two preregistered studies, we assessed whether humorous entertainment education formats on organ donation elicit positive effects on knowledge, fears, attitudes, and behavioral intentions–both immediately after the treatment and four weeks later. We test whether perceived funniness mediates expected effects on attitudes and intentions. Study 1 is a quasi-experiment which uses a live medical comedy show (N = 3,964) as an entertainment education format, which either contained or did not contain information about organ donation. Study 2, a lab experiment, tests humor's causal effect in a pre-post design with a control group (N = 144) in which the same content was provided in either a humorous or non-humorous way in an audio podcast. Results showed that humorous interventions per se were not more effective than neutral information, but that informing people about organ donation in general increased donation intentions, attitudes, and knowledge. However, humorous interventions were especially effective in reducing fears related to organ donation. The findings are discussed regarding the opportunities for sensitive health communication through entertainment education formats, psychological processes that humor triggers, and humor's role in health communication formats.

## Introduction

Humor makes us unique. We laugh 17 times a day on average [1], and by doing so, we immediately feel better. These positive physiological effects happen unconsciously [2] and serve various purposes, such as providing relief from tension, resolving incongruity, or expressing superiority. Humor is defined as an emotional appeal based on a positive emotion, leading to "heightened arousal, smile, and laughter exhibited by an audience in response to a particular message" [3]. Advertisements [4] and health communications [5] use humor strategically to

**Funding:** The author(s) received no specific funding for this work.

**Competing interests:** The authors have declared that no competing interests exist.

raise awareness. This work assesses the potential of humorous entertainment education formats, such as comedy shows [6] or entertainment podcasts, to increase knowledge and change attitudes and intentions in the sensitive field of organ donation.

## Humor in entertainment education formats

*Entertainment education* is defined as the "intentional placement of educational content in entertainment messages"(Singhal, Cody, Rogers, & Sabido, 2004, p. 117), comprising a broad range of formats, such as movies, TV programs, videos, pop music, spectator sports, theme parks, radio, casinos, magazines, newspapers, books, and toys [7].

A systematic review of humor in health communication [8], which included 12 eligible full texts (from 1,451 identified publications) revealed that in previous studies, humor often was an element of entertainment education approaches [9] and can increase involvement and identification, which can influence the audience's attitudes and intentions [10]. While humorous communication should not be expected to change actual health behavior, it may influence persuasion and behavioral intentions [11]. A meta-analysis revealed that humor increases messages' persuasive power ($r = .35$, $p < .01$) and is related to attitude changes ($r = .38$, $p < .01$) [12]. However, when considering only experimental studies (that can draw causal conclusions), no advantage was found from using humorous messages compared with serious messages [13–16], except in Yoon [17]. The review also revealed a lack of experimental designs with a clear theoretical foundation and replications of findings.

## Humor's psychological mechanisms as message strategy

One of the most frequently used models for explaining humor's effects is the Elaboration Likelihood Model (ELM) [4]. As it also has been proposed to explain entertainment education formats' processes [18], we chose this model as the present studies' basis. The ELM assumes that given high motivation and ability to process information (e.g., due to high issue involvement), information is processed via the central route [19,20]. Proposed arguments are analyzed carefully, argument strength determines attitude change, and attitude change is rather stable. Given low involvement, processing via the peripheral route takes place. In this case, one uses less cognitive effort, and attitude change is based on simple cues and heuristics, such as warm feelings toward the communicator or trust in an expert [21]. Attitude change via the peripheral route is less stable than the central route.

Evidence concerning the processing of humor in health messages is mixed. Bae (2008) argues that in humorous contexts, recipients develop an affective response to the communicator and, therefore, have increased issue involvement, which reinforces cognitive elaboration of the message–proposing the central persuasion route. In contrast, Young [22] finds elaboration of humor to be highly complex and cognitively demanding, leaving no cognitive capacity for defense mechanisms such as reactance or counter-arguing [5,23,24]. Therefore, it is proposed that processing takes place via the peripheral route. Thus, while both [25] and Young [22] expect humor to lead to attitude change, the assumed processes differ. Following these research lines, we will interpret humor's positive effects on attitude change's stability as indicators of central route processing, and less reactance and counter-arguing as indicators of peripheral processing.

## Organ donation as an important health communication domain

The current study assesses whether humor is an effective health communication strategy in the organ donation domain. Previous studies have used entertainment education formats in this domain [10,25–27], yet none of the studies tested experimentally whether humorous messages

can increase willingness to carry an organ donation card or positively influence attitudes and behavior. In Germany, for example, more than 10,000 people waited for a donated organ in 2017 [28]. While most Germans (84%) hold positive attitudes toward organ donation, only half the adult population has made a donation decision, and only one-fifth has documented such decisions through organ donation cards or living wills [29]. As people frequently state that they lack information about organ donation and mention fear as a major reason for opposing organ donation [29], entertainment education formats should aim at effectively providing relevant knowledge and decrease fears. Therefore, next to attitudes and intentions, knowledge and fears are used as dependent variables in the present studies as well.

## Hypotheses

Based on the aforementioned elaborations, the present studies test the following general preregistered hypotheses (https://osf.io/vkn4s).

**Entertainment education hypothesis.**   After people are exposed to an entertaining education format about organ donation, we expect more positive attitudes and intentions related to organ donation, compared with respective *a priori* measures. Moreover, we expect that the changes occur due to the information given, i.e., a control group without such input will not show these changes. We also expect greater knowledge and fewer fears after an entertaining education format about organ donation.

**Humor hypothesis.**   We expect that humorous information will lead to greater changes in knowledge, fears, attitudes, intentions, and behavior than non-humorous information, and we assume that the subjective perception of humor (funniness) mediates this relation (Study 2).

**Peripheral processing hypothesis.**   We will test whether perceived funniness is related to less reactance and counter-arguing, mediating humor-induced changes in attitudes and intentions (Study 2), as well as indicating peripheral information processing.

We will explore whether humor's effects on attitudes and intentions are stable over time and more stable than changes resulting from non-humorous interventions. This will challenge the peripheral processing hypothesis, as stable changes point toward central processing [25].

## Overview

In two studies, we assessed whether humorous entertainment education formats containing information about organ donation exert positive effects on knowledge, fears, attitudes, and behavioral intentions–both immediately after the intervention and after four weeks. Study 1's field setting is a live medical comedy show, i.e., an entertainment education format, in which participants viewed either an episode that contained humorous information about organ donations, or one without it. Study 2 tested humor's causal effects in a pre-post lab experiment with a control group, in which the same information was provided either in a humorous or non-humorous way in an audio podcast.

# Study 1

This quasi-experiment was conducted as a paper-and-pencil study in two live episodes of a medical comedy show.

## Method

The episodes took place in Erfurt, Germany (intervention group: included 10 minutes on organ donation) and in Leipzig (control group: organ donation not mentioned) in April 2018. The show's host was Dr. Eckart von Hirschhausen, a famous German physician, comedian,

TV host, and science journalist. He developed the medical comedy show "*Endlich*" (a German pun, meaning both "at last" and "finite") to inform and entertain the audience on different topics related to finiteness, e.g., death, getting older, and activities to prolong life (e.g., healthy eating, physical activity, quitting smoking).

**Ethical considerations**. The study is negligible risk research (no foreseeable risk of harm or discomfort; and any foreseeable risk is no more than inconvenience) and (b) it involves only non-identifiable data about human beings [30]. Participation was voluntary and non-participation was possible without any consequences at all time points. The data were analyzed anonymously. Negligible risk research is exempt from IRB approval.

**Participants and design.** All participants were part of the audience of the live show. They had shown up to view the live show and did not know that the show included a study. The viewers were invited to voluntarily participate in the study. The quasi-experiment implemented a 2 (treatment: intervention vs. control) x 3 (time: T1 [before the show]; T2 [before the show's intermission]; and T3 [after four weeks]) mixed factorial design. Data from T1 and T2 were collected via written questionnaires, while T3 data were collected online. A self-generated personal code linked data anonymously. Those who participated at all three time points received a free audiobook as an incentive. All available data were used, but due to missing data, the sample sizes vary in the analyses. Of the $N = 3,964$ participants ($n_{\text{intervention}} = 2,896$, $n_{\text{control}} = 1,068$) who were included in T1/T2 analyses, 56% were female; mean age was $M = 50.76$, $SD = 14.60$; and 37% held a university degree. For $N = 513$ participants, we had data from all three data assessments (S1 Fig). Before the show, about 45% already had made an organ donation decision, 34% had filled out an organ donation card, and 41% had communicated their decisions to family or friends.

**Treatment.** During the show, the host either included or did not include humorous 10 minutes on organ donation immediately before the intermission. In the piece, the aim was to transfer knowledge humorously and debunk different fears that people face when thinking about organ donation (audio file https://osf.io/vkn4s).

**Measures.** The original questionnaires are available at https://osf.io/vkn4s. Table 1 provides an overview of all measures, sample items, references, their quality indicators (Cronbach's alpha) and whether it was measured before (T1) and/or after the intervention (T2) and/or after four weeks (T3). At T1, we assessed demographics, participants' previous behavior, and baseline attitudes and intentions. All dependent variables (attitudes, intentions, knowledge, fears) related to organ donation were measured immediately post-treatment (T2) and four weeks later (T3). At T2, we measured all items related to the treatment (perceived funniness, reactance, counter-arguing) and immediate behavior (taking the organ donation card home). Participants received either the knowledge or fear items to reduce the time to complete the questionnaire. The questionnaire at T3 contained additional questions about participants' behavior since participating in the study and whether they further engaged in the topic of organ donation.

**Procedure.** The host instructed the audience to fill out the questionnaires placed on their seats at T1 (before the show) and at T2 (before the intermission, which was immediately after the treatment in the treatment condition). Questionnaires were returned in an envelope, and contact details for T3 were collected separately to guarantee anonymity. The participants were free to take a leaflet with an organ donation card with them, which was attached to the envelope, but they were not actively invited to do so. After four weeks, participants received a link to the online T3 questionnaire, with a debriefing via email sent out a few weeks later.

**Attitudes and intentions.** Preregistered repeated-measures 2 (time) x 2 (treatment) ANOVAs tested the entertainment education hypothesis positing that after the treatment, more positive attitudes and intentions related to organ donation will be present, compared

**Table 1. Measures used in both studies with sample items, answer formats, and quality indicators (where applicable).**

| Construct | Sample item and source | Answer format | Number of items, format, quality indicators | Used in Study 1 | Used in Study 2 |
|---|---|---|---|---|---|
| **Attitude (affective, cognitive, and general attitude)** | Affective: "According to my spontaneous gut instinct, organ donation is. . ." Cognitive: "Considering the advantages and disadvantages of organ donation, I think organ donation is. . ." General: "Now we want to know your personal evaluation of the topic organ donation. How do you evaluate organ donation in general?" [31] | Affective and cognitive: Seven-point semantic differential scales, with "I don't know" as an additional option (bad-good, unimportant-important, unsafe-safe) General: seven-point Likert scale (very negative/very positive) with "I don't know" as an additional option | Seven items; mean across seven items Cronbach's α's between .850 and .915 | T1, T2, T3 | T1, T2, T3 |
| **Knowledge** | "A brain-dead person can awaken again." Based on [32–34] | true, false, I don't know | Seven items (all of which referred to topics mentioned during the treatment); sum score, transformed into POMP score | T2, T3 | T1, T2, T3 |
| **Fears** | "I fear that I'm not really dead after a brain death." Based on [32–34] | Seven-point Likert scales (strongly disagree/strongly agree) | Seven items (all of which referred to topics mentioned during the treatment); sum score, recoded, transformed into POMP score | T2, T3 | T1, T2, T3 |
| **Behavioral intentions** | "In the near future, I intend to": (1) make a decision about organ donation (2) communicate the decision to my family or friends (3) sign an organ donation card [31] | Seven-point Likert scales (extremely unlikely/extremely likely) with an extra answer option (already done) | Three items; mean across three items, Cronbach's α's between 0.903 and 0.932 | T1, T2, T3 | T1, T2, T3 |
| **Previous behavior** | Whether participants already: (1) made a decision (2) communicated their decision to others (3) noted their decision on an organ donation card | Yes, no, I don't know | Three items; Cronbach's α = .897 (Study 1); α = .878 (Study 2) | T1 | T1 |
| **Immediate behavior** | Participants took the organ donation card available in the experiment with them | yes, no | | T2 | T2 |
| **Follow-up behavior** | Participants indicated whether, during the past four weeks, they had: (1) made a decision (2) communicated their decision to others (3) recorded their decision on an organ donation card | Yes, no, I don't know, already done | Three items; Cronbach's α = .838 (Study 1); α = .855 (Study 2) | T3 | T3 |
| **Perceived funniness overall, treatment** | "How funny did you perceive the show in general?" "How funny did you perceive the organ donation stand-up?" (intervention group only) [16] | 10-point Likert scale (not at all funny/extremely funny) | Single items | T2 | |
| **Perceived funniness treatment** | Rating of how information about organ donation was presented | Seven-point semantic differentials (not funny-funny, not amusing-amusing, not humorous-humorous, not entertaining-entertaining) | Four items; mean across four items, Cronbach's α = .949 | | T2 |
| **Reactance** | "I felt a resistance inside because of a strong perceived manipulation" [35] | Seven-point Likert scale (strongly disagree/strongly agree) | Three items; mean across three items, Cronbach's α = .826 (Study 1); α = .825 (Study 2) | T2 | T2 |
| **Counter-arguing** | "I questioned some statements" [23] | Seven-point Likert scale (strongly disagree/strongly agree) | One item* | T2 | T2 |
| **Issue involvement** | "Organ donation is a personally relevant topic for me" [36] | Seven-point Likert scale (strongly disagree/strongly agree), with "I don't know" as an extra option | Seven items; mean across seven items, Cronbach's α = .879 | | T1 |

All questionnaires containing all measures are provided at https://osf.io/vkn4s. POMP = percent of maximum possible score ([observed score-minimum score]/[maximum score-minimum score] x 100). An increase of 1 on a POMP scale corresponds to an increase of 1% on the original scale.

* As the two assessed items were not reliable, we chose the items with greater variance.

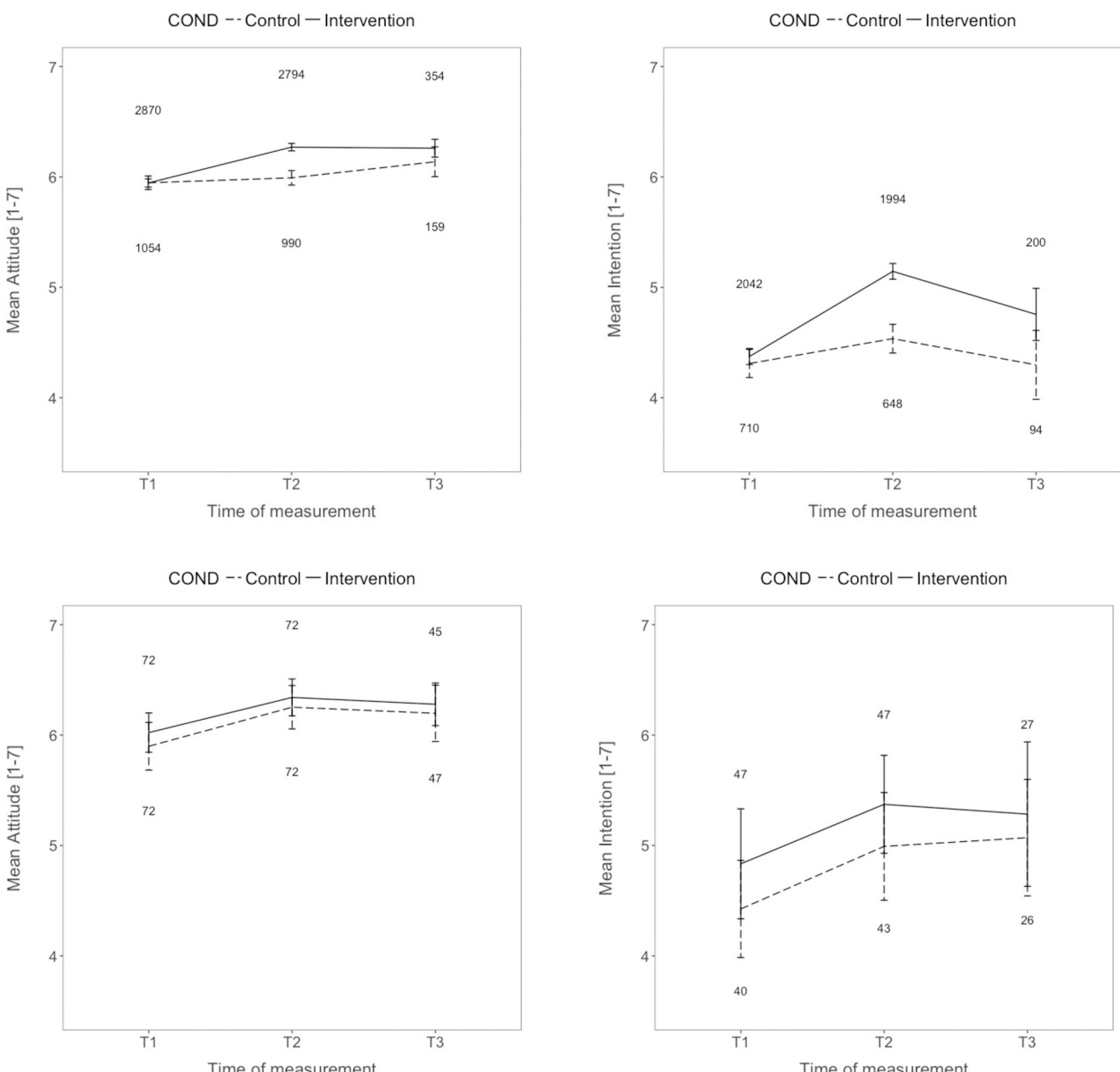

**Fig 1. Mean attitudes (left) and intentions (right) to perform pro-organ donation behaviors across time and as a function of the humorous treatment for Study 1 (top) and Study 2 (bottom).** In Study 1 (top), the humorous treatment positively affected attitudes and intentions compared with no treatment (T1 before the show, T2 immediately after the treatment), but the effect declined after four weeks (T3). In Study 2, the humorous treatment exerted similar positive effects as the neutral control treatment that delivered the same information about organ donation. Error bars are 95% confidence intervals. Numbers indicate n per group. Note that the y axis is cropped (range 1–7).

with *a priori* measures, and that these changes occur only in the intervention group. Fig 1's top panel displays the results with additional explorative data for the four-week follow-up (T3).

For both dependent variables, significant main effects were found both for time (attitude: $F(1, 3,771) = 182.51$, $p < .001$; partial $\text{eta}_p^2 = 0.05$ intention: $F(1, 2,561) = 326.56$, $p < .001$, $\text{eta}_p^2 = 0.11$) as well as for treatment (attitude: $F(1, 3,771) = 16.36$, $p < .001$, $\text{eta}_p^2 = 004.$; intention: $F(1, 2,561) = 21.74$, $p < .001$, $\text{eta}_p^2 = 0.01$). The main effect was qualified by the

A humor hypothesis

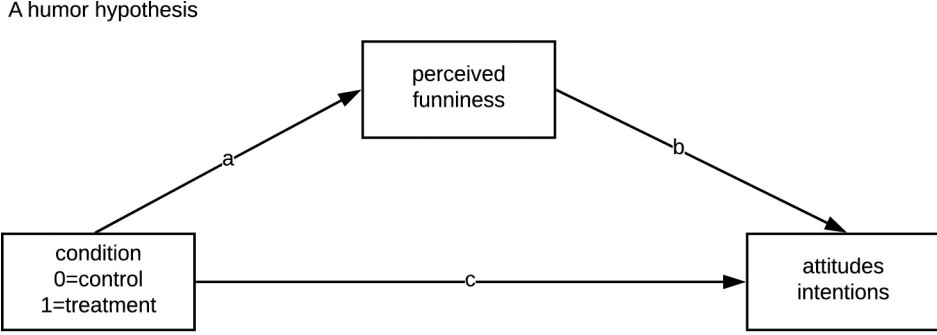

B peripheral processing hypothesis

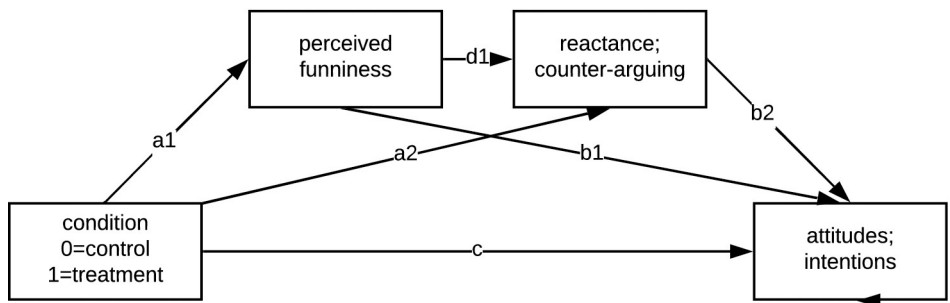

**Fig 2. Schematic mediation models.** Preregistered models tested the humor and peripheral processing hypotheses. The indirect effects of interest are a*b and a*d*b. The numbers correspond to specific coefficients. All mediation results are reported in detail in the Supplement. Models are based on PROCESS (A: Model 4, B: Model 6) [37].

predicted-interaction effect (attitude: $F(1, 3,771) = 136.27$, $p < .001$, $eta_p^2 = 0.04$; intention: $F(1, 2,561) = 94.08$, $p < .001$, $eta_p^2 = 0.04$), indicating that the increase in attitudes and intentions was especially pronounced in the treatment condition. As can be inferred from the 95% confidence intervals (CIs) in Fig 1, attitudes and intentions at T3 still were significantly more positive compared with T1 in the treatment group, while no such substantial and stable change was found in the control group.

As the humor hypothesis proposed, attitudes and intentions at T2 were related to perceived funniness of the treatment (attitudes: $r = .25$ ($p < .001$; n = 2,566); intentions: $r = .27$ ($p < .001$; n = 1,827); note that this was assessed for the treatment group only as the treatment was evaluated which was absent in the control group). Because of the different age groups in the sample, we checked whether age correlated with the perception of funniness in the treatment group. There was a significant correlation with a very small strength of association, $r = .054$, $p = .006$.

In two pre-registered mediation analyses (Fig 2A; PROCESS v3.0 by Hayes (2013), Model 4, using 5,000 bootstrap samples for bias correction; results S1 and S2 Tables), we further tested whether the perceived funniness of the whole show (of which the treatment was only a small part) mediated the treatment's effects on attitudes and intentions. While participants in the treatment condition perceived the whole show as significantly less funny (attitude model: a = -.56, 95%CI [-.68, -.45]; intention model: a = -.60, 95%CI [-.75, -.46]), perceiving the show as funnier led to more positive attitudes (b = .11, [.09, .13]) and intentions (b = .19 [.15, .23]). Therefore, resulting significant indirect effects were negative (attitude: ab = -.06, [-.08, -.05]; intentions: ab = -.11, [-.16, -.08]), indicating that perceiving the whole show as less humorous decreased the treatment's effect. The pattern remained stable when controlling for *a priori* attitudes and intentions, respectively (S3 and S4 Tables).

**Knowledge and fears.** The entertainment education hypothesis also posits that after the treatment, more knowledge and fewer fears will be present than in the control group. Knowledge and fears were assessed at T2 (during the show) and T3 (four weeks afterward). We conducted the two 2 (time: T2, T3) x 2 (condition: treatment, control) preregistered repeated-measures ANOVAs (one for knowledge, one for fears) to assess stability over time. Both analyses revealed the predicted main effects for condition, indicating more knowledge/fewer fears after the treatment (knowledge: $F(1, 232) = 46.39$, $p < .001$, $\text{eta}_p^2 = 0.17$; $F(1, 273) = 4.06$, $p < .05$, partial $\text{eta}^2 = 0.02$). Knowledge declined over time, but fears did not change over time (knowledge: $F(1, 232) = 14.84$, $p < .001$, $\text{eta}_p^2 = 0.06$; fears: $F<1$). No significant interaction effects were found ($F$s$<1$).

**Reactance and counter-arguing.** The peripheral processing hypothesis expects lower reactance and counter-arguing as a consequence of humor-induced peripheral processing. Indeed, statistically significant negative correlations between perceived funniness during the treatment and reactance ($r = -.19$, $p < .001$, n = 2,474), as well as counter-arguing ($r = -.08$, $p < .001$, n = 2,347), were found.

**Behavior.** Regarding immediate behavior, we found that in the intervention group, significantly more people took the organ donation leaflet with them (58%) than in the control group (50%) ($\chi^2$ (1, $N = 3,964$) = 21.26, $p < .001$). Analyzing only those who did not have an organ donation card before the show, the results show the same pattern (60% vs. 53%, respectively). Note that the pre-registered mediation analysis to assess whether the intention to fill out the ODC mediates the effect of condition on behavior was not conducted, as the PROCESS macro cannot process dichotomous dependent variables. Among those who did not have an organ donation card before the show and who received the treatment ($n = 1,880$), an explorative binary logistic regression showed that perceived funniness during the treatment significantly affected the probability of taking the organ donation card home ($B = 0.10$, $SE = 0.03$, $p = 0.001$, Exp B = 1.10). After four weeks, the results showed that those who had perceived the treatment as more humorous filled out the organ donation card significantly more often ($B = 0.23$, $SE = 0.11$, $p = 0.04$, Exp B = 1.25).

## Study 2

While Study 1 had the great advantage of assessing a live comedy show's effects on the audience in a field setting, the study also has several limitations, and parts of the hypotheses could not be tested with the field setting and the resulting design. First of all, participation in the T1/T2 questionnaire in Study 1 was very high, as it was included in a show and the host provided time to participate. Consequently, barriers to participate were low. In contrast, the T3 data were collected via an online questionnaire four weeks later, and barriers for participation were a lot higher (including the willingness to share an e-mail address during the show, having an internet-ready device to participate, mismatches of the T2/T3 codes, etc.). Thus, non-participation at T3 was quite considerable and the explorative analyses regarding T3 data should be interpreted with caution. Moreover, it is not clear whether receiving any information about organ donation or the humorous nature of the message was effective. Further, random assignment of participants to the intervention and control groups was not possible and we cannot exclude the possibility of a-priori differences in knowledge and fears. Finally, it could not be controlled whether participants filled out the questionnaires completely by themselves.

These limitations were addressed in Study 2 using a controlled lab setting that compared a humorous podcast's effects with those of a neutral podcast, with both containing the same information about organ donation. Four weeks later, the participants received the online questionnaire via email. Again, all hypotheses and corresponding tests were pre-registered; data and materials can be accessed at https://osf.io/vkn4s.

## Method

**Statistical power.** *A priori* analysis via G*Power 3.1 [38] was conducted for all hypotheses, resulting in a minimum sample size of $N = 116$ for 1-β = .95 (effect size $f$ = .15; two groups between, three within subjects). Additionally, 20% of the participants were expected to be excluded due to preregistered exclusion criteria; therefore, the recruited sample size increased to $N = 144$.

**Participants and design.** Participants were recruited via ORSEE [39] and participated in exchange for course credit. Within the online questionnaire administered in the Erfurt Laboratory for Empirical Research, the program randomized participants to conditions of the first factor of the 2 (treatment: humorous vs. neutral control, between subjects) x 3 (time: T1 [before podcast], T2 [after podcast], T3 [after four weeks], within subjects) mixed factorial design. The sample size was $N = 144$ ($n_{\text{treatment}} = 72$, $n_{\text{control}} = 72$ at T1, T2), 113 of which were female (79%), with respondents' mean age at 21.03 ($SD = 2.47$). All participants held a high school diploma qualifying for university admission or a higher education level, 54% had already made a decision about organ donation before the study, 51% already filled out an organ donation card, and 44% had already communicated their decisions to family or friends.

**Ethical considerations.** The study is negligible risk research (no foreseeable risk of harm or discomfort; and any foreseeable risk is no more than inconvenience) and (b) it involves only non-identifiable data about human beings [30]. Participation was voluntary and non-participation was possible without any consequences at all time points. The data were analyzed anonymously. Written informed consent was collected from all participants. Negligible risk research is exempt from IRB approval.

**Treatment.** Both groups listened to a podcast with auxiliary graphic materials presented on a computer screen. The information provided was equal in both the humorous and control podcasts, with only the way in which the information was provided varying, leading to differences in time (7:33 minutes [treatment] vs. 4:08 minutes [control]). The podcast (available at https://osf.io/vkn4s) was presented by the same comedian from Study 1.

**Measures.** Table 1 provides an overview of the measures and when they were assessed (T1-T3). The same variables as those in Study 1 were assessed repeatedly to assess changes due to the treatment.

**Procedure.** At T1 and T2, each participant was placed in a cubicle and used a computer to answer the questionnaire at his or her own pace. The audio podcast was provided via headset. Three organ donation leaflets were placed in each cubicle containing organ donation cards. Thus, participants were allowed to take one without being explicitly asked to do so. After four weeks, the participants received an email with a link to the T3 online questionnaire.

## Results

**Manipulation check.** Humor was manipulated successfully, as the humorous podcast led to significantly more perceived funniness than the neutral podcast ($M_{\text{treatment}} = 5.24$ ($SD = 1.37$), $M_{\text{control}} = 2.29$ ($SD = 0.97$), $F(1, 142) = 222.59$, $p < .001$, $\text{eta}_{\text{p}}^2 = .61$). Further, participants' age did not correlate with the perception of funniness of the treatments, $r = -.022$, $p = .798$.

**Attitudes and intentions.** The humor hypothesis expects that attitudes and intentions will increase to a greater extent at T2 in the treatment than in the control condition. As preregistered, repeated measures ANOVAS were conducted with attitudes and intentions as dependent variables and time (T1 vs. T2) and condition (treatment vs. control) as factors. Fig 1 (lower panel) displays the results with additional explorative data for the four-week follow-up (T3). For both dependent variables, time was the only significant main effect (attitude: $F$[1,

142] = 74.37, $p < .001$; $eta_p^2 = .34$; intention: $F[1, 84] = 24.11$, $p < .001$, $eta_p^2 = .22$), indicating more positive attitudes and intentions after either podcast. All other effects were not significant ($F$s<1). Thus, contradicting the humor hypothesis, no benefit was found from the humorous podcast. As can be inferred from the 95% CIs in Fig 1, attitudes and intentions at T3 still were significantly more positive in both groups compared with T1, but again, no difference was found between the treatment and control conditions.

In the pre-registered mediation analyses (Fig 2A; PROCESS v3.0 by Hayes (2013), Model 4, using 5,000 bootstrap samples for bias correction; S5 and S6 Tables), we further tested whether an indirect effect on attitudes and intentions from the treatment was found via perceived funniness while listening to the podcast. Participants with the humorous treatment perceived the podcast as significantly more humorous (a = 2.94, 95%CI [2.55, 3.33]), but these participants did not show more positive attitudes (b = .10, [-.00, .21]) and intentions (b = .25 [-.01, .50]). Contradicting the humor hypothesis, no indirect effects (attitude: ab = .31, [-.06, .64]; intentions: ab = .71, [-.20, 1.53]) were found. The results remained stable when controlling for T1 attitudes or intentions, respectively (S7 and S8 Tables).

**Knowledge and fears.** Knowledge and fears were assessed both at T1 and T2, and results are displayed in Fig 3 (exploring additionally T3). The entertainment education hypothesis predicted an increase (knowledge)/decrease (fears) from listening to the podcast, while the humor hypothesis expected a stronger change due to the humorous (vs. the control) podcast. The preregistered repeated-measures ANOVA with knowledge as the dependent variable and time (T1 vs. T2) and treatment (treatment vs. control) as factors revealed a significant effect only from time ($F[1, 142] = 340.68$, $p < .001$, $eta_p^2 = .71$), indicating more knowledge after the podcasts. However, no effect for treatment and no significant interaction between time and treatment ($F$s <1) were found, contradicting the humor hypothesis.

We repeated the same analysis with fears as the dependent variable, in which time was a significant factor ($F(1, 142) = 249.01$, $p < .001$, $eta_p^2 = .64$), indicating fewer fears after listening to the podcasts (Fig 3; note that higher values indicate less fear). No main effect was found for treatment ($F < 1$), but the interaction was significant ($F(1, 142) = 4.51$, $p < .05$, $eta_p^2 = .03$), indicating that fears decreased more strongly after the humorous (vs. the control) podcast. In sum, the evidence for the humor hypothesis was mixed, i.e., while humor did not affect attitudes, knowledge, and intentions, it slightly lowered fears related to organ donation.

**Reactance and counter-arguing.** The peripheral processing hypothesis expects lower reactance and counter-arguing as a consequence of humor-induced peripheral processing. Therefore, four additional preregistered mediation models were calculated, expecting that the treatment influences perceived funniness, which will decrease reactance and counter-arguing and, consequently, increase attitudes and intentions, leading to a multiple mediation effect as visualized in Fig 2B (S9–S12 Tables).

The results with reactance as a mediator revealed that while perceived funniness increased through the treatment, perceived funniness did not lower reactance significantly (attitudes: d1 = -.12 [-.29, 0.04]; intentions: d1 = -.11 [-.31, 0.09]). Reactance was related significantly to attitudes (b2 = -.34 [-.43, -.24]) and intentions (b2 = -.46 [-.71, -.21]). However, no indirect effects were found via humor and reactance on attitudes (a1d1b2: .12 [-.03, .29]) and on intentions (a1d1b2: .15 [-.11, .51]).

With counter-arguing as a mediator, the pattern was somewhat different for attitudes and intentions: When attitude was the model's dependent variable, perceived funniness significantly lowered counter-arguing (d1 = -.24 [-.47, -0.01]), and less counter-arguing was related to more positive attitudes (b2 = -.15 [-.22, -0.07]). However, no significant indirect effect was found via humor and counter-arguing on attitudes (a1d1b2: .10 [-.00, .23]). When intention was the dependent variable, none of the effects was significant (d1 = -.24 [-.49, 0.01]; b2 = -.18

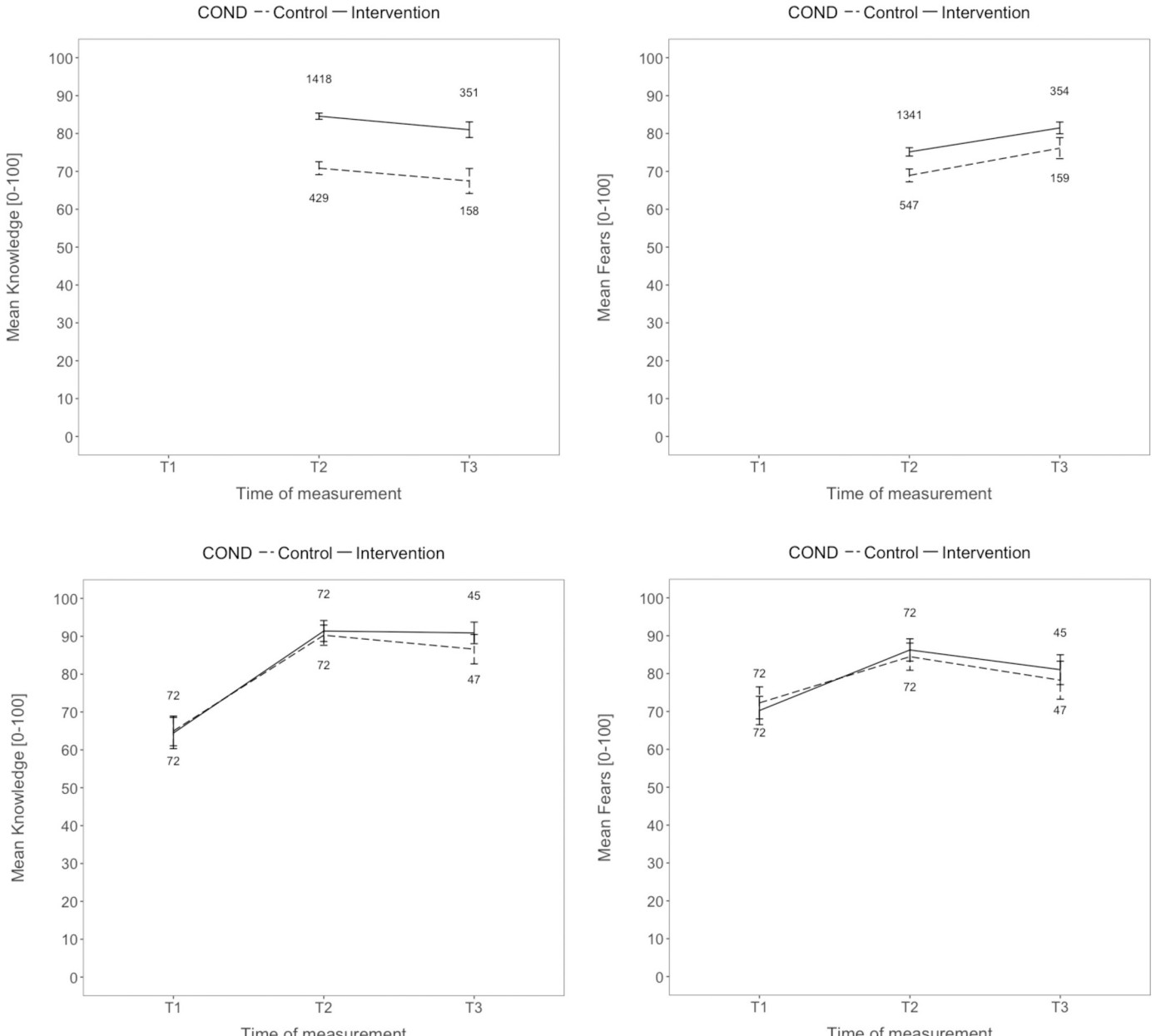

**Fig 3. Mean knowledge (left) and fears (right) regarding organ donation across time and as a function of the humorous treatment for Study 1 (top) and Study 2 (bottom).** Knowledge (sum ranging from 0–7) and fears (mean ranging from 1–7) were transformed into POMP scores (percentage of maximum possible scores, Table 1, ranging from 0–100%). Fear scores were re-coded, so higher scores mean less fear/more correct knowledge. Fear and knowledge items comprise the same content, but were framed either as knowledge or fears. Interestingly, mean percentage correct was higher when the items were framed as knowledge (left) instead of fears (right). Both studies show that humorous interventions lead to fewer fears (right). Error bars are 95% CIs. Numbers indicate n per group.

[-.40, 0.04]; a1d1b2: .12 [-.05, .33]). All analyses indicate the same results pattern when controlled for *a priori* attitudes and intentions (S13–S16 Tables), except for intention as dependent variable (S15 Table).

In sum, the peripheral processing hypothesis was not supported. While perceived funniness was related partially to less counter-arguing and reactance, this still did not lead to changes in the dependent variables as expected.

**Behavior.** Regarding immediate behavior, we found that in both conditions, exactly 20% of participants took the organ donation leaflet with them. As only 43 participants did not have an organ donation card prior to the study, we did not conduct any logistic regressions for T2 and T3 behavior.

**Involvement.** To explore the processes more, we also analyzed the potential effect of involvement. According to the ELM, involvement determines the route of information processing. Particularly participants with low involvement levels are expected to use peripheral processing. If humor is processed via the peripheral route, one would expect an effect from humor only for individuals with low involvement. Thus, we explored in four additional analyses (PROCESS v3.0 by Hayes (2013), Model 1, using 5,000 bootstrap samples for bias correction, S17–S20 Tables) whether *a priori* involvement moderated the effect from the humor treatment or perceived funniness on either attitudes or intentions. In all four analyses, involvement exerted a significant positive main effect on attitudes and intentions, indicating that participants with higher involvement held higher post-treatment attitudes and intentions. This indicates central processing. Correspondingly, none of the moderator effects was significant, contradicting the peripheral processing hypothesis. Analyses that controlled for *a priori* attitudes and intentions showed the same results pattern.

## General discussion

Both studies provide evidence for the entertainment education hypothesis, as both the live medical comedy show and the podcasts indicated positive short-term effects on attitudes and led to more knowledge about organ donation, fewer related fears, and greater intentions to perform pro-organ donation behaviors. While these changes occurred only in the intervention group in Study 1, they occurred after listening to both the humorous and neutral podcasts in Study 2. While the effects declined somewhat over time, in both studies the positive effects from the interventions remained visible after four weeks. Therefore, we conclude that entertainment education approaches are effective means to educate people about organ donation.

Regarding the humor hypothesis, in Study 1, we found that perceiving more funniness during the treatment led to more positive attitudes and greater intentions toward pro-organ donation behaviors, yielding support for the humor hypothesis. The mediation results showing that not perceiving the show as humorous decreased the treatment effect also yields evidence for the humor hypothesis. However, in Study 2, intentions, attitudes, and knowledge did not increase more when the information was provided or perceived in a humorous way. Only fear reduction was related to perceived funniness. It is possible that the effects from humor actually were smaller than expected, and that we, therefore, were unable to detect them with the given sample size. In Study 1, the treatment had an effect of $f = .10$ to $.20$, and in the power analysis for Study 2, we assumed an effect size of $f = .15$. With a smaller expected effect size of $f = .10$, the power analysis would have resulted in a sample size of $N = 260$ for $1-\beta = .95$ (two groups between, three groups within).

As stated before, receiving information at all and humor was confounded, so the actual effect in Study 1 did not only result from humor. Our literature review also showed that previous experimental studies indicated no advantage from humor [13–16], except in Yoon [17]. While larger samples could be used to test for the smaller effect, even small effects may be useful if such interventions are used for large-scale behavioral change. We conclude that humor may contribute to entertainment education formats' positive effect, but may not be the only, nor the strongest, mechanism.

It still is likely that humorous content influences especially affective outcomes. In the case of organ donation, this seems especially valuable as fears are one of the major factors that

prevent people from donating organs [32,40]. This also may contribute to lowering the threshold for seeking information about the sensitive topic. Thus, future research should explore humor's potential in providing an entry point for people who are reluctant to seek organ donation information.

The evidence regarding processes elicited by humor is mixed. In Study 1, partial support was found for the peripheral processing hypothesis, showing that more perceived funniness was related to less counter-arguing and reactance. The changes in attitudes and intentions also declined somewhat after four weeks, also suggesting peripheral, rather than central, processing. In Study 2, perceived funniness also affected reactance and counter-arguing, but this did not lead to the expected change in attitudes and intentions. Additionally, those who were highly involved in the topic exhibited greater changes in attitudes and intentions, indicating central, rather than peripheral, processing. Thus, in both studies, we found certain attitude-change stability, indicating central route processing, and less reactance and counter-arguing, indicating peripheral processing. One possible interpretation is that both processes were involved and were triggered to different degrees in both studies. Being in a lab in front of a computer for 30 minutes might trigger more controlled processing, while being in an audience for a 120-minute comedy show may lead to more peripheral processing. It is impossible to disentangle this, given the present studies. Future studies should consider that processing humorous messages and information provided in entertainment education formats also may be influenced by the context in which the information is consumed [8]. It is noteworthy that in both studies, reactance was lowered by humor, despite the context being a sensitive, death-related topic.

While the studies included different age groups, both examined mainly highly educated people. Future studies should test whether humorous entertainment education formats also work with less-educated people. The overrepresentation of highly educated people may also explain why the proportion of people with organ donation card was somewhat higher than the German average. All participants were included in the analysis, independent from their prior behavior (e.g., even when they already had an organ donation card). This could have weakened the results and the effect from (humorous) interventions. Yet, repeating the main ANOVAS on attitudes and intentions with only those participants without an organ donation card yielded the same results.

In sum, with this work, we followed the theoretical agenda for entertainment education research put forward by [7]. The studies: (1) went beyond only assessing whether entertainment education exerts an effect and also examined how this effect occurred; (2) broadened the assessed formats' scope by evaluating a medical comedy show's effects [6]; (3) examined recipient-bound resistance to entertainment education effects by assessing reactance [41]; (4) paid attention to affective aspects by making humor a central independent variable; and (5) employed methodological pluralism by using a live show setting and a more controlled lab setting, examining both self-reporting and behavior as dependent variables. The work also builds on existing research (e.g., by Bae (2008) or Bae and Kang (2008) by using (quasi-)experimental longitudinal designs.

## Conclusion

The two studies showed that entertainment education formats are valuable for delivering information about organ donation and can increase knowledge, acting as important determinants behavior. Humor proved useful, especially for reducing fears about organ donation. Importantly, humor did not increase reactance in both studies, but rather reduced it. Thus, even with a sensitive topic, humor seems to be an appropriate strategy and can improve

outcomes from entertainment education interventions. Future interventions should use these findings to improve communication around organ donation, as well as the number of people who decide about and document their decisions on organ donation.

## Supporting information

**S1 Fig. Sample sizes at T1, T2, T3.**
(TIF)

**S1 Table. Mediation analysis: Effect of treatment (X) on attitude T2 (Y) via perceived funniness (M), model 4 (Hayes, 2013).** $n$ = 3,510. Treatment: 0 = control group without topic of organ donation, 1 = intervention group with organ donation stand-up. Attitude: mean across seven items, ranging from 1 to 7. Perceived funniness: 1 = not humorous to 10 = humorous. 95% BC CI: corrected 95% confidence interval with lower and upper border, based on 5,000 bootstrap resamples, CIs that do not contain zero indicate a significant indirect effect with $p <$ .05.
(DOCX)

**S2 Table. Mediation analysis: Effect of treatment (X) on intention T2 (Y) via perceived funniness (M), model 4 (Hayes, 2013).** $n$ = 2,444. Treatment: 0 = control group without topic of organ donation, 1 = intervention group with organ donation stand-up. Intention: mean across three items, ranging from 1 to 7. Perceived funniness: 1 = not humorous to 10 = humorous. 95% BC CI: corrected 95% confidence interval with lower and upper border, based on 5,000 bootstrap resamples, CIs that do not contain zero indicate a significant indirect effect with $p <$ .05.
(DOCX)

**S3 Table. Mediation analysis: Effect of treatment (X) on attitude T2 (Y) via perceived funniness (M), controlled for the attitude T1 (covariate), model 4 (Hayes, 2013).** $n$ = 3,504. Treatment: 0 = control group without topic of organ donation, 1 = intervention group with organ donation stand-up. Attitude: mean across seven items, ranging from 1 to 7. Perceived funniness: 1 = not humorous to 10 = humorous. 95% BC CI: corrected 95% confidence interval with lower and upper border, based on 5,000 bootstrap resamples, CIs that do not contain zero indicate a significant indirect effect with $p <$ .05.
(DOCX)

**S4 Table. Mediation analysis: Effect of treatment (X) on intention T2 (Y) via perceived funniness (M), controlled for the intention T1 (covariate), model 4 (Hayes, 2013).**
$n$ = 2,379. Treatment: 0 = control group without topic of organ donation, 1 = intervention group with organ donation stand-up. Intention: mean across three items, ranging from 1 to 7. Perceived funniness: 1 = not humorous to 10 = humorous. 95% BC CI: corrected 95% confidence interval with lower and upper border, based on 5,000 bootstrap resamples, CIs that do not contain zero indicate a significant indirect effect with $p <$ .05.
(DOCX)

**S5 Table. Mediation analysis: Effect of treatment (X) on attitude T2 (Y) via perceived funniness (M), model 4 (Hayes, 2013).** $n$ = 144. Treatment: 0 = neutral control treatment, 1 = humorous treatment. Attitude: mean across three items, ranging from 1 to 7. Perceived funniness: mean across four items, ranging from 1 to 7. 95% BC CI: corrected 95% confidence interval with lower and upper border, based on 5,000 bootstrap resamples, CIs that do not contain zero indicate a significant indirect effect with $p <$ .05.
(DOCX)

**S6 Table. Mediation analysis: Effect of treatment (X) on intention T2 (Y) via perceived funniness (M), model 4 (Hayes, 2013).** $n$ = 90. Treatment: 0 = neutral control treatment, 1 = humorous treatment. Intention: mean across three items, ranging from 1 to 7. Perceived funniness: mean across four items, ranging from 1 to 7. 95% BC CI: corrected 95% confidence interval with lower and upper border, based on 5,000 bootstrap resamples, CIs that do not contain zero indicate a significant indirect effect with $p < .05$.
(DOCX)

**S7 Table. Mediation analysis: Effect of treatment (X) on attitude T2 (Y) via perceived funniness (M), controlled for the attitude T1 (covariate), model 4 (Hayes, 2013).** $n$ = 144 Treatment: 0 = neutral control treatment, 1 = humorous treatment. Attitude: mean across seven items, ranging from 1 to 7. Perceived funniness: mean across four items, ranging from 1 to 7. 95% BC CI: corrected 95% confidence interval with lower and upper border, based on 5,000 bootstrap resamples, CIs that do not contain zero indicate a significant indirect effect with $p < .05$.
(DOCX)

**S8 Table. Mediation analysis: Effect of treatment (X) on intention T2 (Y) via perceived funniness (M), controlled for the intention T1 (covariate), model 4 (Hayes, 2013).** $n$ = 86. Treatment: 0 = neutral control treatment, 1 = humorous treatment. Intention: mean across three items, ranging from 1 to 7. Perceived funniness: mean across four items, ranging from 1 to 7 95% BC CI: corrected 95% confidence interval with lower and upper border, based on 5,000 bootstrap resamples, CIs that do not contain zero indicate a significant indirect effect with $p < .05$.
(DOCX)

**S9 Table. Mediation analysis: Effect of treatment (X) on attitude T2 (Y) via perceived funniness (M1) and reactance (M2), model 6 (Hayes, 2013).** $n$ = 144. Attitude: mean across seven items, ranging from 1 to 7. Perceived funniness: mean across four items, ranging from 1 to 7. Reactance: mean across three items, ranging from 1 to 7. 95% BC CI: corrected 95% confidence interval with lower and upper border, based on 5,000 bootstrap resamples, CIs that do not contain zero indicate a significant indirect effect with $p < .05$.
(DOCX)

**S10 Table. Mediation analysis: Effect of treatment (X) on attitude T2 (Y) via perceived funniness (M1) and counter-arguing (M2), model 6 (Hayes, 2013).** $n$ = 144. Attitude: mean across seven items, ranging from 1 to 7. Perceived funniness: mean across four items, ranging from 1 to 7. Counter-arguing: single item, ranging from 1 to 7. 95% BC CI: corrected 95% confidence interval with lower and upper border, based on 5,000 bootstrap resamples, CIs that do not contain zero indicate a significant indirect effect with $p < .05$.
(DOCX)

**S11 Table. Mediation analysis: Effect of treatment (X) on intention T2 (Y) via perceived funniness (M1) and reactance (M2), model 6 (Hayes, 2013).** $n$ = 90. Intention: mean across three items, ranging from 1 to 7. Perceived funniness: mean across four items, ranging from 1 to 7. Reactance: mean across three items, ranging from 1 to 7. 95% BC CI: corrected 95% confidence interval with lower and upper border, based on 5,000 bootstrap resamples, CIs that do not contain zero indicate a significant indirect effect with $p < .05$.
(DOCX)

**S12 Table. Mediation analysis: Effect of treatment (X) on intention T2 (Y) via perceived funniness (M1) and counter-arguing (M2), model 6 (Hayes, 2013).** $n$ = 90. Intention: mean across three items, ranging from 1 to 7. Perceived funniness: mean across four items, ranging from 1 to 7. Counter-arguing: single item, ranging from 1 to 7. 95% BC CI: corrected 95% confidence interval with lower and upper border, based on 5,000 bootstrap resamples, CIs that do not contain zero indicate a significant indirect effect with $p < .05$.
(DOCX)

**S13 Table. Mediation analysis: Effect of treatment (X) on attitude T2 (Y) via perceived funniness (M1) and reactance (M2), controlled for the attitude T1 (covariate), model 6 (Hayes, 2013).** $n$ = 144. Attitude: mean across seven items, ranging from 1 to 7. Perceived funniness: mean across four items, ranging from 1 to 7. Reactance: mean across three items, ranging from 1 to 7. 95% BC CI: corrected 95% confidence interval with lower and upper border, based on 5,000 bootstrap resamples, CIs that do not contain zero indicate a significant indirect effect with $p < .05$.
(DOCX)

**S14 Table. Mediation analysis: Effect of treatment (X) on attitude T2 (Y) via perceived funniness (M1) and counter-arguing (M2), controlled for the attitude T1 (covariate), model 6 (Hayes, 2013).** $n$ = 144. Attitude: mean across seven items, ranging from 1 to 7. Perceived funniness: mean across four items, ranging from 1 to 7. Counter-arguing: single item, ranging from 1 to 7. 95% BC CI: corrected 95% confidence interval with lower and upper border, based on 5,000 bootstrap resamples, CIs that do not contain zero indicate a significant indirect effect with $p < .05$.
(DOCX)

**S15 Table. Mediation analysis: Effect of treatment (X) on intention T2 (Y) via perceived funniness (M1) and reactance (M2), controlled for the intention T1 (covariate), model 6 (Hayes, 2013).** $n$ = 86. Intention: mean across three items, ranging from 1 to 7. Perceived funniness: mean across four items, ranging from 1 to 7. Reactance: mean across three items, ranging from 1 to 7. 95% BC CI: corrected 95% confidence interval with lower and upper border, based on 5,000 bootstrap resamples, CIs that do not contain zero indicate a significant indirect effect with $p < .05$.
(DOCX)

**S16 Table. Mediation analysis: Effect of treatment (X) on intention T2 (Y) via perceived funniness (M1) and counter-arguing (M2), controlled for the intention T1 (covariate), model 6 (Hayes, 2013).** $n$ = 86. Intention: mean across three items, ranging from 1 to 7. Perceived funniness: mean across four items, ranging from 1 to 7. Counter-arguing: single item, ranging from 1 to 7. 95% BC CI: corrected 95% confidence interval with lower and upper border, based on 5,000 bootstrap resamples, CIs that do not contain zero indicate a significant indirect effect with $p < .05$.
(DOCX)

**S17 Table. Moderation analysis: Effect of treatment (X) on attitude T2 (Y) moderated by involvement (W), model 1 (Hayes, 2013).** $n$ = 144. Treatment: 0 = neutral control treatment, 1 = humorous treatment. Attitude: mean across seven items, ranging from 1 to 7. Involvement: mean across seven items, ranging from 1 to 7. 95% CI: 95% confidence interval with lower and upper border, CIs that do not contain zero indicate a significant indirect effect with $p < .05$.
(DOCX)

**S18 Table. Moderation analysis: effect of perceived humour (X) on attitude T2 (Y) moderated by involvement (W), model 1 (Hayes, 2013).** $n$ = 144. Perceived funniness: mean across four items, ranging from 1 to 7. Attitude: mean across seven items, ranging from 1 to 7. Involvement: mean across seven items, ranging from 1 to 7. 95% CI: 95% confidence interval with lower and upper border, CIs that do not contain zero indicate a significant indirect effect with $p < .05$.
(DOCX)

**S19 Table. Moderation analysis: Effect of treatment (X) on intention T2 (Y) moderated by involvement (W), model 1 (Hayes, 2013).** $n$ = 90. Treatment: 0 = neutral control treatment, 1 = humorous treatment. Intention: mean across three items, ranging from 1 to 7. Involvement: mean across seven items, ranging from 1 to 7. 95% CI: 95% confidence interval with lower and upper border, CIs that do not contain zero indicate a significant indirect effect with $p < .05$.
(DOCX)

**S20 Table. Moderation analysis: Effect of perceived humour (X) on intention T2 (Y) moderated by involvement (W), model 1 (Hayes, 2013).** $n$ = 90. Perceived funniness: mean across four items, ranging from 1 to 7. Intention: mean across three items, ranging from 1 to 7. Involvement: mean across seven items, ranging from 1 to 7. 95% CI: 95% confidence interval with lower and upper border, CIs that do not contain zero indicate a significant indirect effect with $p < .05$.
(DOCX)

## Acknowledgments

The reported studies were part of Dominik Daube, Sarah Drexler, Tetyana Kotovnykova, Nora Katharina Küpke, Leonie Otten, and Karen Pauli's research internships to gain course credit for the health communication master's program at the University of Erfurt. Their great efforts in collecting, entering, and analyzing data, as well as conducting a comprehensive literature review, are gratefully acknowledged. The original report is available at https://osf.io/vkn4s.

## Author Contributions

**Conceptualization:** Cornelia Betsch, Nora K. Schmid-Küpke, Leonie Otten, Eckart von Hirschhausen.

**Data curation:** Cornelia Betsch, Nora K. Schmid-Küpke, Leonie Otten.

**Formal analysis:** Nora K. Schmid-Küpke, Leonie Otten.

**Investigation:** Nora K. Schmid-Küpke, Leonie Otten.

**Methodology:** Cornelia Betsch, Nora K. Schmid-Küpke, Leonie Otten.

**Resources:** Eckart von Hirschhausen.

**Supervision:** Cornelia Betsch.

**Visualization:** Cornelia Betsch.

**Writing – original draft:** Cornelia Betsch, Nora K. Schmid-Küpke, Leonie Otten.

**Writing – review & editing:** Cornelia Betsch, Nora K. Schmid-Küpke, Leonie Otten, Eckart von Hirschhausen.

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
