## [Decision Letter · Decision Letter 0]

23 Sep 2020

PONE-D-20-21037

Increasing the Willingness to Participate in Organ Donation Through Humorous Health Communication: (Quasi-)Experimental Evidence

PLOS ONE

Dear Dr. Betsch,

Thank you for submitting your manuscript to PLOS ONE. After careful consideration, we feel that it has merit but does not fully meet PLOS ONE’s publication criteria as it currently stands. Therefore, we invite you to submit a revised version of the manuscript that addresses the points raised during the review process.

We look forward to receiving your revised manuscript.

Kind regards,

Valerio Capraro

Academic Editor

PLOS ONE

Additional Editor Comments:

I have now collected one review from one expert in the field. The review is extremely positive and suggests acceptance conditional to a minor revision. After reading the paper, I agree with the reviewer. Therefore, I would like to invite you to revise your work following their comments. I am looking forward for the final version.

Journal Requirements:

Reviewers' comments:

Reviewer's Responses to Questions

**Comments to the Author**

1. Is the manuscript technically sound, and do the data support the conclusions?

Reviewer #1: Yes

2. Has the statistical analysis been performed appropriately and rigorously? 

Reviewer #1: Yes

3. Have the authors made all data underlying the findings in their manuscript fully available?

Reviewer #1: Yes

4. Is the manuscript presented in an intelligible fashion and written in standard English?

Reviewer #1: Yes

5. Review Comments to the Author

Reviewer #1: This is a well planned, executed, and presented paper, examining the role of Entertainment education, and more specifically humour, in eliciting positive attitudes on knowledge, fear, attitudes, and behavioural intentions regarding organ donation. Specific strengths of the paper:

1) The methodology and design of the study is based on thorough systematic literature review of the field.

2) Considerable thought has been given to statistical power and sample size calculation for study 2

3) Appropriate use of statistical tests and relevant inferences drawn from the results

Some minor suggestions are mentioned below:

1) Typographical error - Study 1, attitude and intentions section, 3rd paragraph......the line reads "intentions at r = .27 (p < 0.001)"

2) Study 1 and 2 included significantly different age groups of populations, and whether age has any correlations with perception of funniness of the treatments, and thereby an effect on attitude and intention changes, is something that may be touched upon in the discussion.

3) Although the authors have discussed the role of low statistical power in their non obtaining of a significant effect of humour on attitude, intention, knowledge in study 2 (general discussion paragraph 2), perhaps a word or two regarding the same in terms of a post-hoc power analysis with calculated effect sizes may be considered.

6. PLOS authors have the option to publish the peer review history of their article (what does this mean?). If published, this will include your full peer review and any attached files.

Reviewer #1: **Yes: **Akshay T Jagadeesh

---

## [Author Response · Author response to Decision Letter 0]

9 Oct 2020

In addition to the three changes mentioned below we changed the OSF view only link to the full access link: https://osf.io/vkn4s. We further changed the formatting according to the PLOS ONE requirements. 

R1: 1) Typographical error - Study 1, attitude and intentions section, 3rd paragraph......the line reads "intentions at r = .27 (p < 0.001)"

Response: We have changed the text into: “As the humor hypothesis proposed, attitudes and intentions at T2 were related to perceived funniness of the treatment (attitudes: r = .25 (p <.001; n=2,566); intentions: r =.27 (p <.001; n =1,827); note that this was assessed for the treatment group only as the treatment was evaluated which was absent in the control group).”

R1: 2) Study 1 and 2 included significantly different age groups of populations, and whether age has any correlations with perception of funniness of the treatments, and thereby an effect on attitude and intention changes, is something that may be touched upon in the discussion.

Response: Thank you for this suggestion. We explored the correlations as suggested. We added the results to the results section of Study 1: (p. 10): “Because of the different age groups in the sample, we checked whether age correlated with the perception of funniness in the treatment group. There was a significant correlation with a very small strength of association, r = .054, p = .006.” In the results section of Study 2 we added: “Further, participants’ age did not correlate with the perception of funniness of the treatments, r = -.022, p = .798.” As the correlation was very low and did not appear in the second study, we refrained from adding this to the discussion. 

R1: 3) Although the authors have discussed the role of low statistical power in their non obtaining of a significant effect of humour on attitude, intention, knowledge in study 2 (general discussion paragraph 2), perhaps a word or two regarding the same in terms of a post-hoc power analysis with calculated effect sizes may be considered.

Response: Thank you for this suggestion. As it is suggested “explain how likely it was to observe a significant effect, given your sample, and given an expected or small effect size” instead of calculating post-hoc power analyses (http://daniellakens.blogspot.com/2014/12/observed-power-and-what-to-do-if-your.html) we added: “With a smaller expected effect size of f = .10, the power analysis would have resulted in a sample size of N = 260 for 1-�= .95 (two groups between, three groups within).”

---

## [Editor Report · Decision Letter 1]

12 Oct 2020

Increasing the Willingness to Participate in Organ Donation Through Humorous Health Communication: (Quasi-)Experimental Evidence

PONE-D-20-21037R1

Dear Dr. Betsch,

We’re pleased to inform you that your manuscript has been judged scientifically suitable for publication and will be formally accepted for publication once it meets all outstanding technical requirements.

Kind regards,

Valerio Capraro

Academic Editor

PLOS ONE
---

## [Editor Report · Acceptance letter]

19 Oct 2020

PONE-D-20-21037R1 

Increasing the Willingness to Participate in Organ Donation Through Humorous Health Communication: (Quasi-)Experimental Evidence 

Dear Dr. Betsch:

I'm pleased to inform you that your manuscript has been deemed suitable for publication in PLOS ONE. Congratulations! Your manuscript is now with our production department. 

Kind regards, 

on behalf of

Dr. Valerio Capraro 

Academic Editor

PLOS ONE